# Fecal Transplantation from db/db Mice Treated with Sodium Butyrate Attenuates Ischemic Stroke Injury

Huidi Wang,[a,b] Wei Song,[a] Qiheng Wu,[a] Xuxuan Gao,[b] Jie Li,[b] Chuhong Tan,[a] Hongwei Zhou,[b] Jiajia Zhu,[a] Yan He,[b] Jia Yin[a]

[a]Department of Neurology, Nanfang Hospital, Southern Medical University, Guangzhou, Guangdong, China
[b]Microbiome Medicine Center, Department of Laboratory Medicine, Zhujiang Hospital, Southern Medical University, Guangzhou, Guangdong, China

Huidi Wang, Wei Song, and Qiheng Wu contributed equally to this article. Author order was determined in order of increasing seniority.

**ABSTRACT** The complication of type 2 diabetes (T2D) exacerbates brain infarction in acute ischemic stroke (AIS). Because butyrate-producing bacteria are decreased in T2D and butyrate has been reported to be associated with attenuated brain injury in AIS, we hypothesize that administering butyrate could ameliorate T2D-associated exacerbation of brain infarction in AIS. Therefore, we first validated that Chinese AIS patients with T2D comorbidity have significantly lower levels of fecal butyrate-producing bacteria and butyrate than AIS patients without T2D. Then, we performed a 4-week intervention in T2D mice receiving either sodium butyrate (SB) or sodium chloride (NaCl) and found that SB improved the diabetic phenotype, altered the gut microbiota, and ameliorated brain injury after stroke. Fecal samples were collected from T2D mice after SB or NaCl treatment and were transplanted into antibiotic-treated C57BL/6 mice. After 2 weeks of transplantation, the gut microbiota profile and butyrate level of recipient mice were tested, and then the recipient mice were subjected to ischemic stroke. Stroke mice that received gut microbiota from SB-treated mice had a smaller cerebral infarct volume than mice that received gut microbiota from NaCl-treated mice. This protection was also associated with improvements in gut barrier function, reduced serum levels of lipopolysaccharide (LPS), LPS binding protein (LBP), and proinflammatory cytokines, and improvements in the blood-brain barrier.

**IMPORTANCE** Ischemic stroke is a major global health burden, and T2D is a well-known comorbidity that aggravates brain injury after ischemic stroke. However, the underlying mechanism by which T2D exacerbates stroke injury has not been completely elucidated. A large amount of evidence suggests that the gut microbiota composition affects stroke outcomes. Our results showed that the gut microbiota of T2D aggravated brain injury after ischemic stroke and could be modified by SB to afford neuroprotection against stroke injury. These findings suggest that supplementation with SB is a potential therapeutic strategy for T2D patients with ischemic stroke.

**KEYWORDS** type 2 diabetes mellitus, ischemic stroke, sodium butyrate, gut microbiota, fecal microbiota transplantation

Address correspondence to Jiajia Zhu, 32102002@qq.com, Yan He, bioyanhe@gmail.com, or Jia Yin, yinj@smu.edu.cn.

Stroke is the second leading cause of death from cardiovascular disease, accounting for 17% of total deaths (1). Type 2 diabetes (T2D) predisposes humans to acute ischemic stroke (AIS) and is associated with poorer stroke outcomes, including worse neurological outcomes and higher mortality, readmission, and stroke recurrence rates (2, 3). Studies using murine models of these diseases have further confirmed this conclusion (4, 5). However, the underlying mechanism by which T2D exacerbates stroke injury has not been completely elucidated.

In recent years, researchers have paid increasing attention to the role of the gut microbiota in cardiovascular disease and metabolic disorders (6, 7). Studies have

revealed the alterations in the gut microbiota that occur in T2D; meanwhile, a consensus concerning which bacteria are altered significantly in individuals with T2D is lacking, but a common observation is that the abundance of butyrate-producing bacteria is decreased in individuals with this condition (8–10). Butyrate, a short-chain fatty acid (SCFA) that is naturally produced by the bacterial fermentation of complex fiber in the colon (11), shows promising effects on obesity, diabetes, and neurological disorders. Indeed, increased fiber consumption or butyrate supplementation has been shown to decease adiposity and improve insulin sensitivity (12–14). Moreover, butyrate increases mitochondrial-dependent oxygen consumption in enterocytes, stabilizes the hypoxia-inducible factor (HIF) that is involved in barrier protection, and upregulates the expression of HIF target genes that increase barrier function (15).

It has been reported that the gut microbiota composition may affect stroke outcomes. Our previous work showed that dysbiosis of the gut microbiota is correlated with a worse stroke outcome in patients (16). Recently, we reported that ischemic stroke triggers gut microbiota dysbiosis, which in turn exacerbates brain infarction (17). Similar findings have also been documented in other studies. Singh et al. (18) observed an altered microbiota profile after cerebral ischemia, and poststroke dysbiosis is associated with the induction of the inflammatory response. Fecal microbiota transplantation (FMT) of a balanced microbiota after cerebral ischemia improves stroke outcomes. Benakis et al. (19) discovered that a gut microbiota disturbance leads to the exacerbation of ischemic brain lesions by increasing the number of intestinal proinflammatory T cells. Spychala et al. (20) showed that transplantation of the gut microbiota from young mice improves stroke outcomes in aged mice, suggesting that strategies targeting the microbiota composition might exert beneficial effects on individuals with a high risk of stroke.

Due to single-gene mutations that lead to deactivation by the cognate receptor of the satiety factor leptin, db/db mice spontaneously develop severe hyperphagia, leading to obesity and the manifestation of some T2D-like characteristics. Using db/db mice, we examined the effects of sodium butyrate (SB) on diabetes-related parameters, the gut microbiota profile, and stroke injury of T2D mice. Then, we subjected antibiotic-treated mice to FMT from diabetic (Db) mice or wild-type (WT) mice treated with or without SB. Using a middle cerebral artery occlusion (MCAO) model of stroke, we showed that the gut microbiota in individuals with T2D is causally linked to the exacerbation of brain damage and can be modified by SB to afford neuroprotection against acute ischemic stroke injury.

## RESULTS

**General characteristics and gut microbiota profile of study participants.** In the present study, 37 patients with both AIS and T2D and 37 age- and sex-matched patients with AIS but without T2D were recruited. As shown by the data in Table 1, compared to non-T2D patients, patients with T2D presented with a more severe stroke (assessed using the National Institutes of Health Stroke Scale [NIHSS]). We studied the $\alpha$- and $\beta$-diversity and compared the microbial diversity within and between communities. We did not observe differences in $\alpha$- or $\beta$-diversity between the two groups (Fig. 1a and b). Next, we calculated the percentages of bacterial taxa in each group at both the phylum and family levels (Fig. 1c), and then we applied a linear discriminant analysis effect size (LEfSe) method and identified several taxa with significantly different abundances. Notably, butyrate-producing bacteria, such as *Lachnospira*, *Blautia*, and *Butyricicoccus*, were enriched in non-T2D patients compared with their levels in patients with T2D (Fig. 1d). Consistently, the fecal butyrate level was higher in non-T2D patients than in patients with T2D (Fig. 1e). Moreover, serum levels of lipopolysaccharide (LPS) and D-lactate (DLA) were significantly increased in patients with T2D (Fig. 1f and g), indicating the presence of a leaky gut in patients with T2D after AIS.

**Sodium butyrate improves diabetes-related parameters and attenuates stroke injury.** After acclimatization for 1 week, WT and Db mice were randomized to receive either sodium butyrate (SB) or NaCl (as a control [Con]) in the drinking water for

**TABLE 1** Baseline characteristics of the patients with AIS presenting with or without T2D

| | No. (%) or median value (IQR) for patients with: | | |
|---|---|---|---|
| Characteristic | Both AIS and T2D ($n$ = 37) | AIS but without T2D ($n$ = 37) | $P$ value |
| Age (yr) | 61 (19.5) | 62 (17.5) | 0.75 |
| Male | 21 (56.8) | 23 (62.2) | 0.64 |
| History of smoking | 13 (35.1) | 10 (27.0) | 0.45 |
| History of hypertension | 29 (78.3) | 23 (62.2) | 0.13 |
| NIHSS score at admission | 5 (7.5) | 3 (7.5) | 0.044 |
| Large artery atherosclerosis | 21 (56.8) | 20 (54.1) | 0.82 |
| Small vessel occlusion | 11 (29.7) | 10 (27.0) | 0.80 |
| Cardioembolism | 5 (13.5) | 7 (18.9) | 0.53 |
| RBG (mmol/liter) | 10.04 (4.90) | 6.13 (0.81) | <0.001 |
| FBG (mmol/liter) | 7.55 (2.77) | 5.70 (1.02) | <0.001 |
| HbA1c (%) | 7.4 (3.0) | 5.6 (0.8) | <0.001 |
| Body wt (kg) | 64.2 (21.0) | 62.5 (15.5) | 0.096 |
| BMI (kg/m$^2$) | 24.3 (5.3) | 23.2 (3.5) | 0.025 |

4 weeks, and then the mice were subjected to MCAO (Fig. 2a). As expected, during the intervention, the Db mice gained more weight and consumed more food and water than the WT mice, exhibiting hyperglycemia and hyperlipemia (Fig. 2b and c). SB had no effect on body weight, but it significantly reduced the water and food intake of Db mice at week 2 and week 3 of treatment, respectively (Fig. 2b). Moreover, SB reduced fasting blood glucose at weeks 2 and 4 and reduced triglyceride (TG) levels, but it had no significant effect on total cholesterol (T-CHO) levels after treatment (Fig. 2c). At the end of the 4-week intervention period, lower serum levels of LPS, LPS binding protein (LBP), and the proinflammatory cytokines interleukin-6 (IL-6), tumor necrosis factor alpha (TNF-$\alpha$), and IL-1$\beta$ were detected in SB-treated Db mice (Db-SB) than in NaCl-treated Db mice (Db-Con) (Fig. 2d and e). After MCAO, the Db mice exhibited significantly larger infarct volumes than the WT mice. SB treatment had no effect on the infarct volume in WT mice, whereas it significantly reduced the infarct volume in Db mice (Fig. 2f). Significant negative correlations were noted between fecal butyrate and LPS and between fecal butyrate and infarct volume (Fig. 2g). We further analyzed the tight-junction proteins in the gut and observed higher expression of ZO-1, occludin, and claudin-4 in Db-SB mice than in Db-Con mice (Fig. 2h).

**Effects of SB on the gut microbiota profile and SCFA concentrations.** The Db-Con mice had a lower Chao1 index (a proxy for community richness) than WT-Con mice. However, SB treatment did not affect the Chao1 index within either the WT or the Db group (Fig. 3a). Principal-coordinate analysis (PCoA) showed that the gut bacterial characteristics of Db mice were clearly separated from those of WT mice. Additionally, a significant difference in the gut microbiota compositions was observed between the Db groups but not between the WT groups (Fig. 3b). An analysis of the taxonomic composition revealed that Db-SB mice had relatively higher abundances of butyrate-producing bacteria, including *Lachnospiraceae*, *Ruminococcaceae*, *Oscillospira*, and *Ruminococcus*, than Db-Con mice (Fig. 3c). The LEfSe analysis identified several taxa, including *Enterococcus* and *Christensenellaceae*, present at significantly higher abundances in Db-SB mice than in Db-Con mice (Fig. 3d). Furthermore, the fecal concentrations of acetate, propionate, butyrate, valerate, and total SCFAs were tested. The Db mice exhibited consistently lower concentrations of SCFAs than the WT mice. SB treatment significantly increased the fecal butyrate concentration in Db mice (Fig. 3e).

**Gut microbiota profiles and fecal butyrate concentrations in recipient mice after FMT.** We depleted the microbiota in recipient (rWT and rDb) mice by administering a cocktail of antibiotics for 14 days and found that $\alpha$- and $\beta$-diversity were drastically altered by the antibiotic treatment (Fig. S1a and b in the supplemental material). Then, the recipient mice were orally inoculated with fecal suspensions from donor mice once daily for 2 weeks (Fig. 4a). FMT had no effect on diabetes-related parameters in the 4 groups (Fig. S1c and d). The Chao1 index of $\alpha$-diversity was decreased in mice

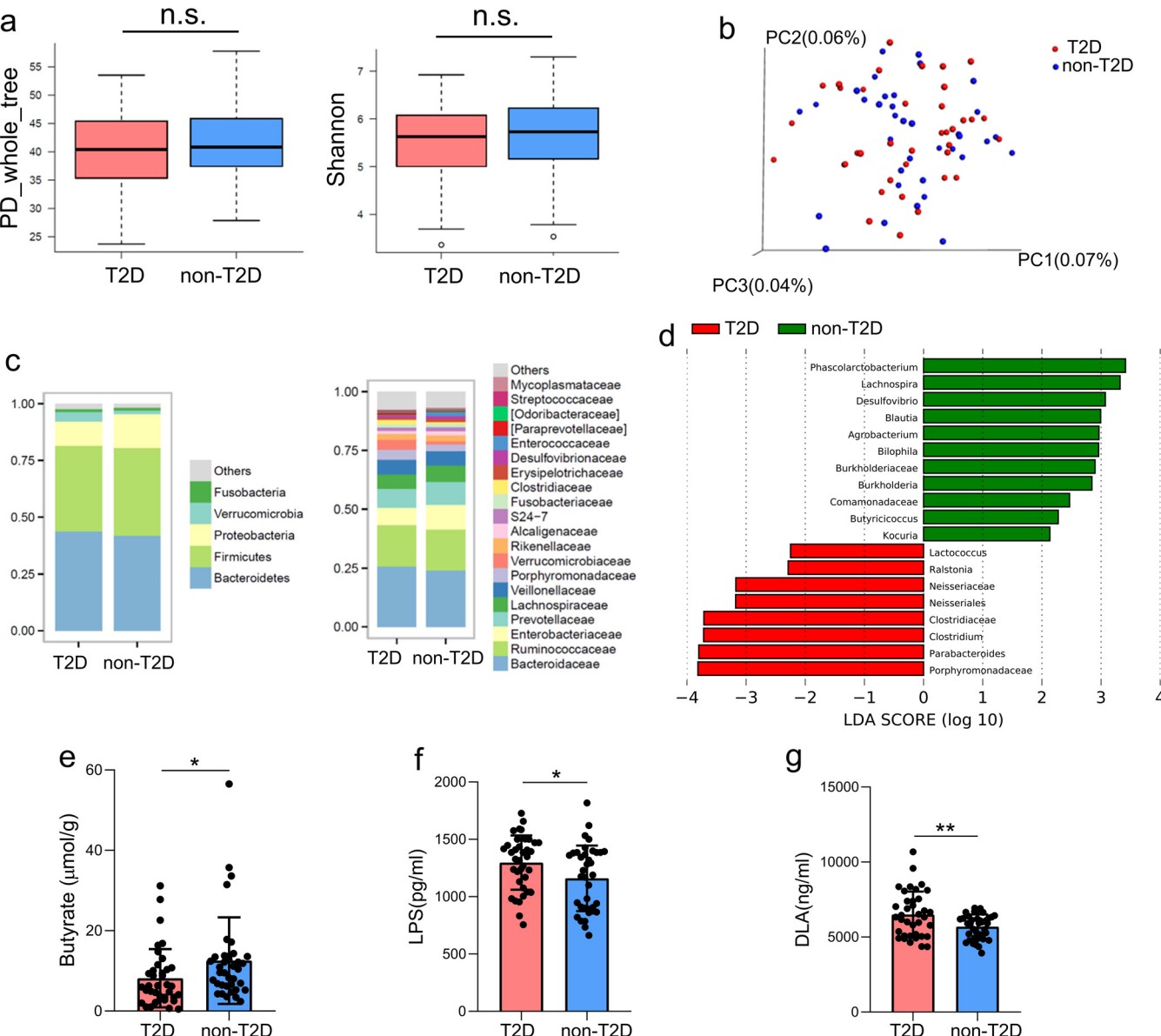

**FIG 1** Gut microbiota, fecal butyrate, and plasma LPS and DLA levels in patients with AIS who presented with or without T2D. (a) $\alpha$-Diversity calculated using PD whole-tree and Shannon indices in patients with AIS who presented with or without T2D. (b) PCoA of $\beta$-diversity based on unweighted UniFrac distances showing the gut microbiota composition among patients with AIS who presented with or without T2D. (c) Average relative abundances of prevalent microbiota at the phylum and family levels in the two groups. (d) LEfSe showing bacterial taxa with significantly different abundances between the two groups. (e to g) Comparison of fecal butyrate levels (e) and plasma LPS (f) and DLA (g) levels between the two groups. $n$ = 37 patients per group. Data are presented as mean values ± SD. *, $P < 0.05$; **, $P < 0.01$.

that received fecal microbiota from NaCl-treated Db mice (rDb-Con) (Fig. 4b). PCoA revealed significant separation of two clusters of rWT and rDb groups, with the gut microbiome of rDb-Con mice differing markedly from that of rDb-SB mice (Fig. 4c). Moreover, the abundances of butyrate-producing bacteria, such as *Oscillospira*, *Lachnospiraceae*, *Ruminococcaceae*, and *Ruminococcus*, were higher in rDb-SB mice than in rDb-Con mice (Fig. 4d to f). We examined the fecal SCFAs in the recipient mice and observed higher fecal butyrate concentrations in rDb-SB mice than in rDb-Con mice (Fig. 4e and Fig. S2).

**Fecal transplantation from db/db mice treated with SB attenuates stroke injury and gut barrier destruction.** Following FMT, the recipient mice were subjected to MCAO for 1 h, followed by reperfusion for 24 h to investigate whether the gut

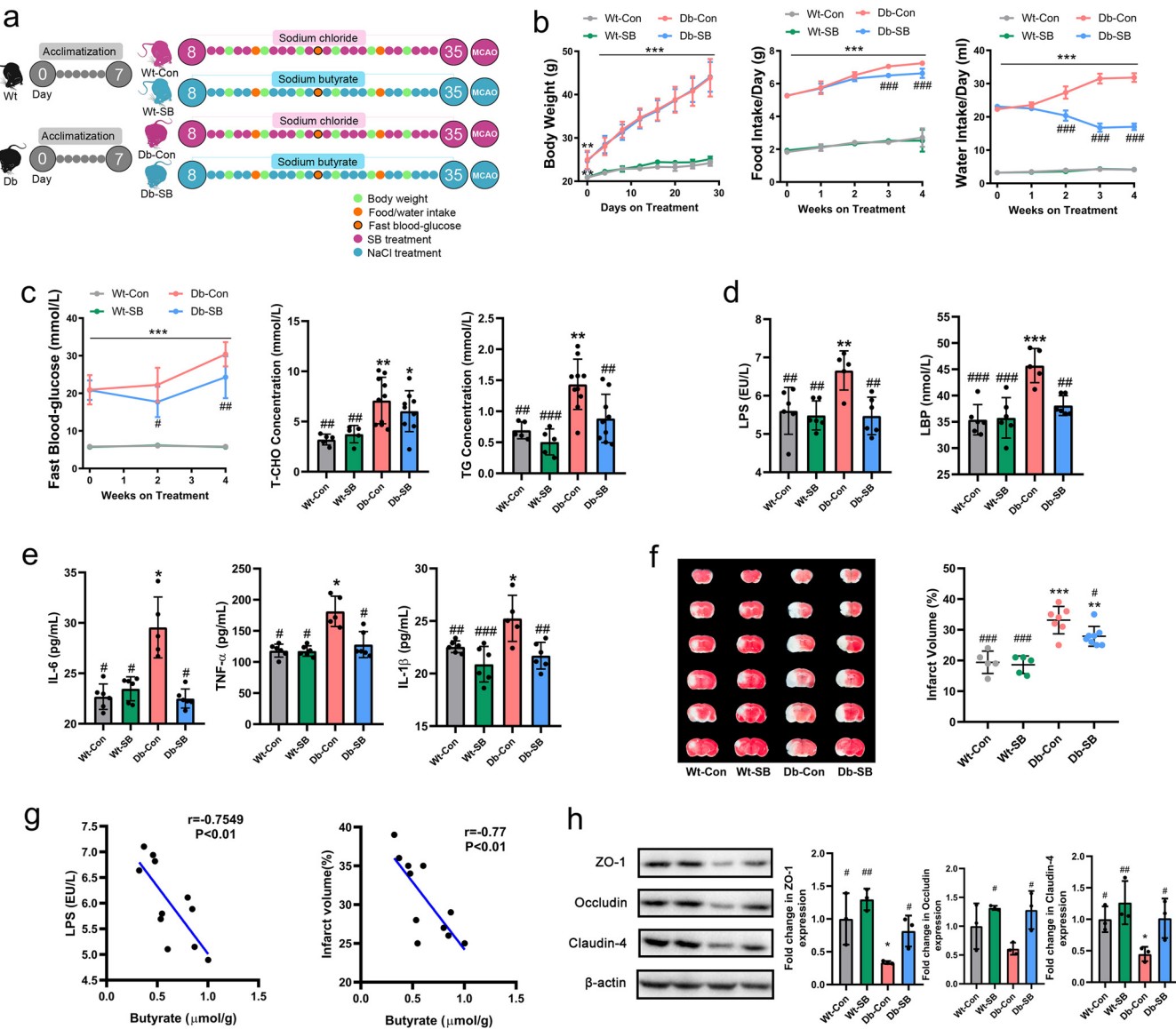

**FIG 2** SB improves diabetes-related parameters and attenuates stroke injury. (a) Experimental design for donor mice. After acclimatization for 1 week, WT and Db mice received NaCl or SB in the drinking water for 4 weeks. Diabetes-related parameters were measured at different time points, and then the mice were subjected to MCAO. (b) Changes in body weight, food intake, and water intake were measured at different time points. (c) Fasting blood glucose levels, T-CHO levels, and TG levels were also measured. (d and e) Serum levels of LPS, LBP (d), IL-6, TNF-$\alpha$, and IL-1$\beta$ (e) were measured after MCAO. (f) Images of brain sections and cerebral infarct volumes as percentages for all MCAO groups. (g) Correlations between fecal butyrate and LPS or infarct volume. (h) Expression of intestinal tight junction proteins in the four groups. $n$ = 5 to 10 animals per group. Data are presented as mean values $\pm$ SD. Asterisks (*) show comparisons with the WT-Con group, and pound signs (#) show comparisons with the Db-Con group. *, $P < 0.05$; **, $P < 0.01$; ***, $P < 0.001$; #, $P < 0.05$; ##, $P < 0.01$; ###, $P < 0.001$.

microbiota of T2D plays a vital role in stroke. The results showed that the rDb-SB mice had lower modified neurological severity scores (mNSS) (Fig. 5a) and smaller infarct volumes (Fig. 5b) than the rDb-Con mice. Interestingly, the effects of FMT on stroke injury remained the same in db/db mice as in the recipients (Fig. S3). Nissl staining was applied to investigate morphological changes in neurons, and we discovered that the rDb-Con mice exhibited more neuronal karyopyknosis, chromatolysis, and shrinkage of cell bodies in the hippocampal CA1 region than mice in the other groups (Fig. 5c). Moreover, the rDb-SB group exhibited less neuronal loss (NeuN positive [NeuN+]) and microglial activation (Iba positive [Iba+]) in the hippocampus and less apoptotic cell death (terminal deoxynucleotidyltransferase-mediated dUTP-biotin nick end labeling [TUNEL]) in the cortex than the rDb-Con group after stroke (Fig. 5d). In addition, we

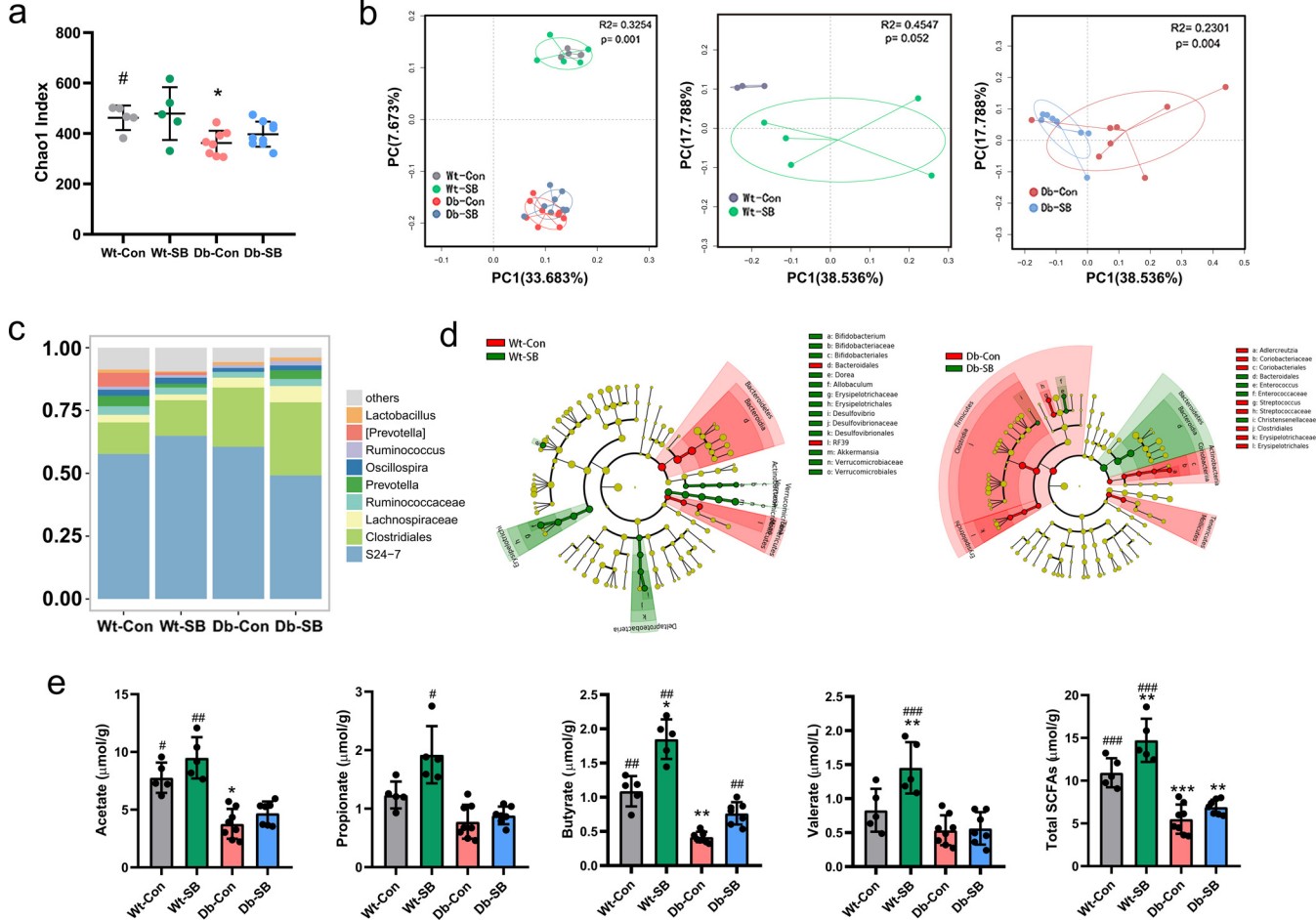

**FIG 3** Effects of SB on the gut microbiota profile and SCFA concentrations. After 4 weeks of intervention, total bacterial DNA was isolated from fecal samples, the 16S rRNA genes were sequenced, and the SCFA concentrations were measured using gas chromatography-mass spectrometry. (a) $\alpha$-Diversity in the four groups measured using Chao-1 index. (b) PCoA plot of unweighted UniFrac distances between the WT and Db groups. Each circle represents a single sample, and circles are color coded based on the group. (c) Average relative abundances of predominant taxa at the genus level in the four groups. (d) LEfSe showing bacterial taxa with significant differences in abundance between groups. (e) Fecal concentrations of SCFAs after 4 weeks of intervention. $n = 3$ to 8 animals per group. Data are presented as mean values $\pm$ SD. Asterisks (*) show comparisons with the WT-Con group, and pound signs (#) show comparisons with the Db-Con group. *, $P < 0.05$; **, $P < 0.01$; ***, $P < 0.001$; #, $P < 0.05$; ##, $P < 0.01$; ###, $P < 0.001$.

assessed intestinal barrier function by analyzing the morphology of the ileum and colon and the expression of barrier-forming tight junction proteins. The rDb-Con mice exhibited a disrupted epithelium with shorter villus height and crypt depth (Fig. 5e) than in the rDb-SB mice. Further analysis revealed higher expression of ZO-1, occludin, and claudin-4 in rDb-SB mice than in rDb-Con mice (Fig. 5f).

**Fecal transplantation from db/db mice treated with SB attenuates ischemic stroke injury by protecting the BBB.** Consistent with the impaired gut barrier of rDb-Con mice, the serum levels of LPS and LBP were higher in rDb-Con mice than in rDb-SB mice, with increased levels of proinflammatory cytokines, including IL-6, TNF-$\alpha$, and IL-1$\beta$ (Fig. 6a). In association with systemic inflammation, the levels of intercellular adhesion molecule-1 (ICAM-1), vascular cell adhesion molecule-1 (VCAM-1), and matrix metalloproteinase-9 (MMP-9) in the brain were markedly higher in rDb-Con mice than in rDb-SB mice (Fig. 6b). The cerebral capillary endothelial glycocalyx is known as the first line of defense protecting the blood-brain barrier (BBB). We assessed the endothelial glycocalyx using electron microscopy and discovered that the rWT mice displayed a thick layer of glycocalyx, whereas only a residual glycocalyx layer was observed in the rDb-Con mice, indicating serious endothelial glycocalyx degradation. On the other hand, the rDb-SB mice had a relatively thicker layer of endothelial glycocalyx than the rDb-Con group (Fig. 6c). Serum levels of syndecan-1 and heparan sulfate (HS) reflect the extent of glycocalyx degradation.

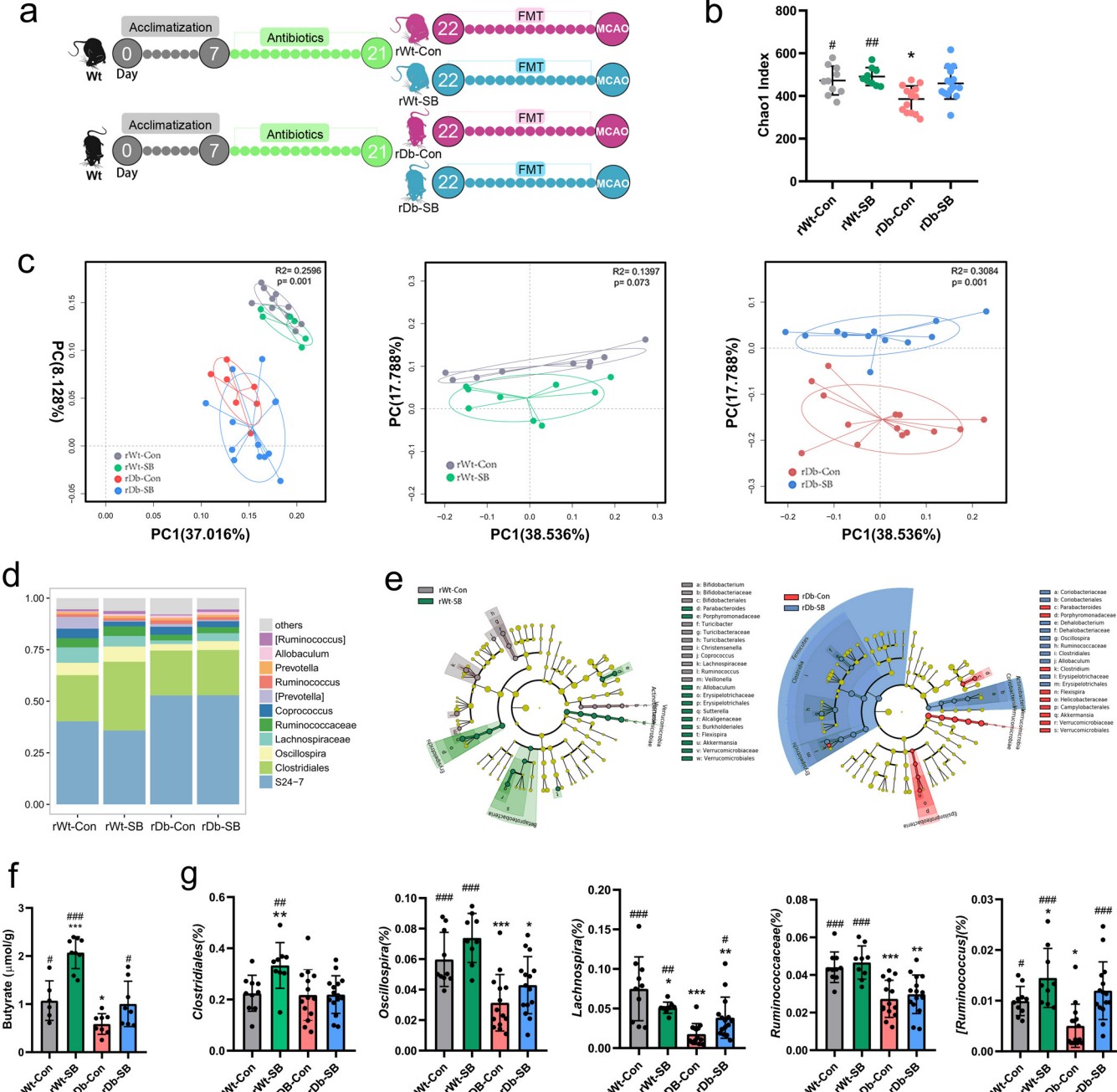

**FIG 4** Gut microbiota profiles and fecal concentrations of butyrate in recipient mice after FMT. (a) Experimental design for recipient mice. After acclimatization, the recipient mice were administered a cocktail of antibiotics for 14 days and then underwent FMT for an additional 14 days. After FMT, the recipient mice were subjected to MCAO for 1 h and sacrificed after 24 h of reperfusion. (b) $\alpha$-Diversity measured using the Chao-1 index in the four groups. (c) PCoA plot of unweighted UniFrac distances between the rWT and rDb groups. (d) Average relative abundances of predominant taxa at the genus level in the four groups. (e) LEfSe showing bacterial taxa with significantly different abundances between groups. (f) Fecal concentrations of butyrate after 2 weeks of FMT. (g) The abundances of butyrate-producing bacteria in the four groups. $n$ = 9 to 15 animals per group. Data are presented as mean values ± SD. Asterisks (*) show comparisons with the rWT-Con group, and pound signs (#) show comparisons with the rDb-Con group. *, $P < 0.05$; **, $P < 0.01$; ***, $P < 0.001$; #, $P < 0.05$; ##, $P < 0.01$; ###, $P < 0.001$.

Consistent with the electron microscopy results, the rDb-Con mice exhibited higher shedding of syndecan-1 and HS than the rDb-SB mice (Fig. 6c). Moreover, the destruction of tight-junction proteins ZO-1, occludin, and claudin-4 was relieved in the rDb-SB mice compared with their levels of destruction in the rDb-Con mice (Fig. 6d).

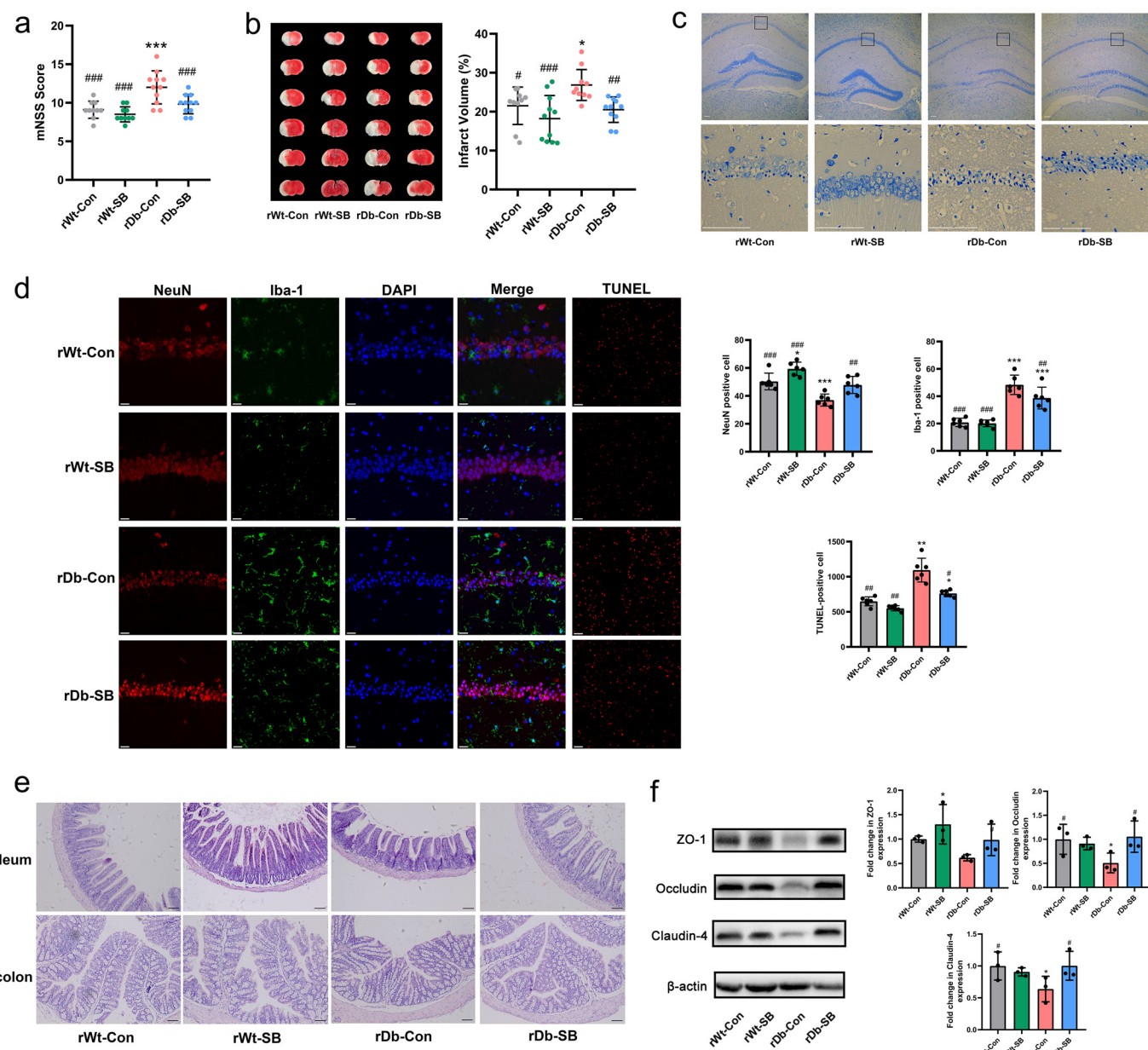

**FIG 5** Fecal transplantation from db/db mice treated with SB attenuates stroke injury and gut barrier destruction. (a) mNSS was assessed before sacrifice. (b) Representative TTC staining of brain slices and percentages of the cerebral infarct volume in the four groups. (c) Representative images of Nissl staining in the hippocampal CA1 region (scale bar = 100 $\mu$m). (d) Double immunostaining for NeuN (neuronal marker) and Iba-1 (microglial marker) was performed in the hippocampal CA1 region, and apoptotic neurons in the peri-infarct cortex were detected using TUNEL staining. DAPI, 4′,6-diamidino-2-phenylindole. (e) The morphology of the ileum and colon was assessed using H&E staining (scale bar = 100 $\mu$m). (f) Expression of intestinal tight junction proteins in the four groups. $n$ = 14 to 16 animals per group. Data are presented as mean values ± SD. Asterisks (*) show comparisons with the rWT-Con group, and pound signs (#) show comparisons with the rDb-Con group. *, $P < 0.05$; **, $P < 0.01$; ***, $P < 0.001$; #, $P < 0.05$; ##, $P < 0.01$; ###, $P < 0.001$.

## DISCUSSION

T2D is a well-known comorbidity that aggravates brain injury after ischemic stroke. The pathophysiological mechanisms by which T2D exacerbates brain injury after stroke have not been completely elucidated. In recent years, tremendous progress has been achieved in revealing the bidirectional interactions that occur between the gut and the brain, namely, the gut-brain axis. Therefore, manipulating or shaping the gut microbiota is attracting attention as a viable strategy to prevent or treat various extra-intestinal diseases.

The primary finding of the present study is that sodium butyrate (SB) modulates the gut microbiota profile of T2D mice and increases the fecal butyrate concentration, which

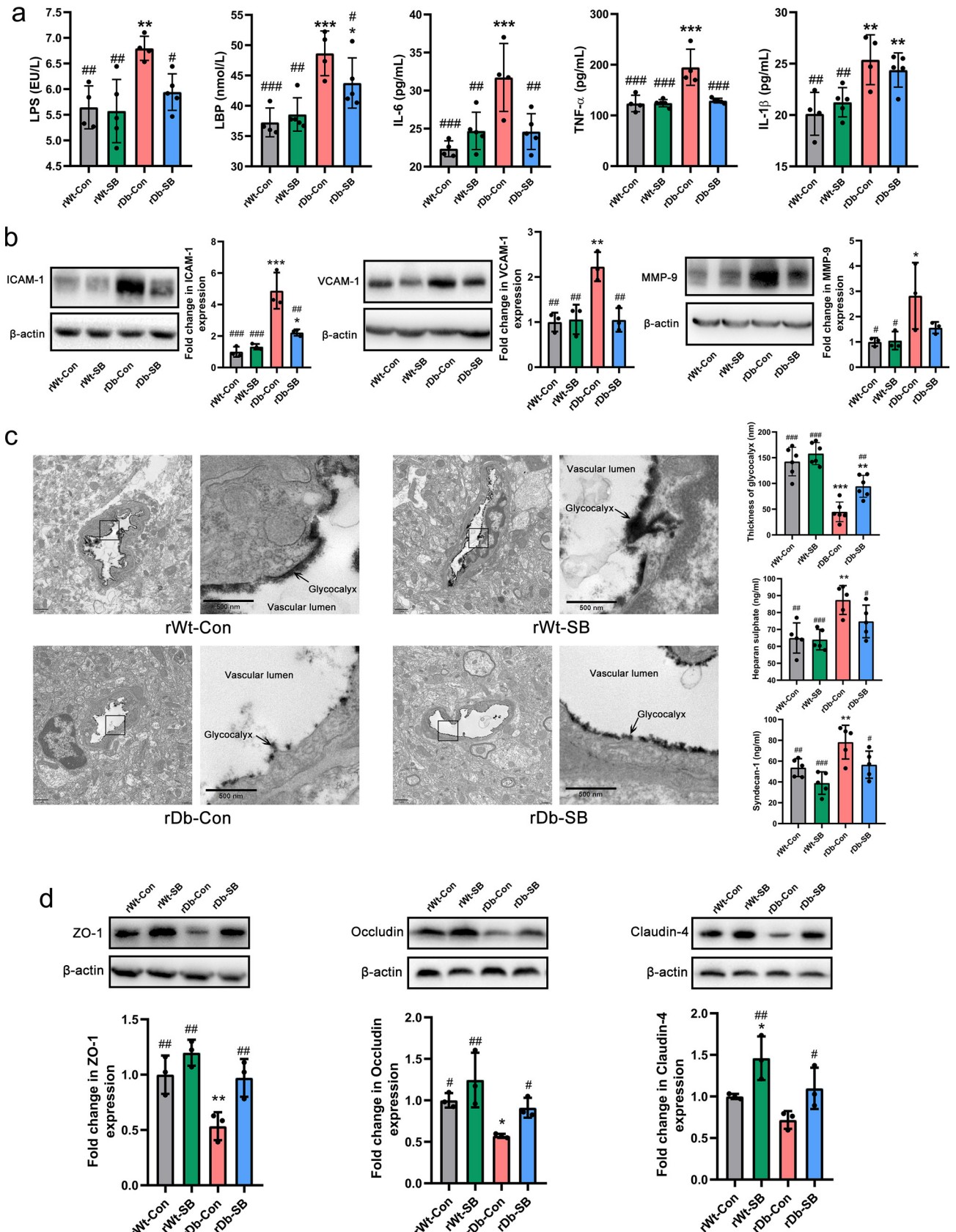

**FIG 6** Fecal transplantation from db/db mice treated with SB attenuates ischemic stroke injury by protecting the BBB. (a) Serum levels of LPS, LBP, IL-6, TNF-$\alpha$, and IL-1$\beta$ in the four groups. (b) Expression of ICAM-1, VCAM-1, and MMP-9 in the brains of the four groups. (c) Electron microscopy

improves stroke outcomes (Fig. 7). This protective effect is related to the enhancement of gut barrier integrity, which translates to reduced serum levels of LPS and proinflammatory cytokines. The alleviated systemic inflammation leads to downregulation of ICAM-1, VCAM-1, and MMP-9 in the brain, which preserves the cerebral endothelial glycocalyx and protects the BBB, resulting in protection against stroke injury.

The gut microbiota dysbiosis in patients with T2D is characterized by reduced levels of many butyrate-producing bacteria (8–10). In addition, plasma LPS and D-lactate (DLA), which are recognized as products and systemic markers of increased gut permeability (21), were increased in patients with T2D. As butyrate is involved in protection of the gut barrier, we speculated that the decreased fecal butyrate level is at least partially responsible for the increased levels of LPS and DLA in patients with T2D. Higher levels of LPS and DLA are associated with worse outcomes in patients with AIS (22). Based on our results, patients with T2D had a higher NIHSS score at admission, which indicates a more severe stroke.

Numerous studies have confirmed the beneficial effects of butyrate on glucose homeostasis and energy homeostasis (23–27). In our study, we discovered that supplementation with butyrate elicited favorable effects on diabetes-related parameters in db/db mice and attenuated stroke injury. A previous study revealed that *Clostridium butyricum* attenuates ischemic stroke injury in diabetic mice by modulating the gut microbiota (28). However, this study does not discount the effect of the blood glucose level on ischemic stroke injury, as treatment with *C. butyricum* significantly alleviates hyperglycemia, which is a strong risk factor for poor stroke outcomes. We applied the FMT technique to test the hypothesis that the T2D gut microbiota *per se* exerts a substantial effect on ischemic stroke injury and to eliminate the possibility that butyrate attenuates stroke injury by exerting metabolic benefits against T2D.

Consistent with a previous study (25), supplementation with SB at a concentration of 0.1 mol/liter for 4 weeks effectively shaped the gut microbiota composition of Db mice in our study. Then, the gut microbiota was transplanted to the recipient mice for a short period of 2 weeks to avoid potential metabolic effects on the recipient mice, because the gut microbiota might be an important contributor to the development of T2D (29, 30). In the present study, we observed that FMT from Db mice had no effect on diabetes-related parameters in recipient mice. Notably, the rDb-Con mice had a significantly higher abundance of *Clostridium* bacteria than the rDb-SB mice. An overgrowth of *Clostridium* is often considered a feature of gut microbiota dysbiosis and increased gut permeability, known as "leaky gut syndrome," a feature of inflammatory bowel disease and other intestinal and extraintestinal immune diseases (31).

After FMT, the recipient mice were subjected to MCAO. We found that the rDb-Con mice exhibited significantly worse performance in terms of the neurological score and had larger infarct volumes than rWT-Con mice, whereas these exacerbations were mitigated in rDb-SB mice. Interestingly, the effects of FMT on stroke injury remained the same in the recipient db/db mice. Moreover, FMT did not influence hyperglycemia in the recipient db/db mice, indicating that the beneficial effect on hyperglycemia in donor Db mice was directly afforded by SB in the drinking water. Notably, the rDb-Con mice suffered more severe gut barrier disruption than rDb-SB mice after stroke. LPS activates gastrointestinal immune cells to release proinflammatory cytokines from the gut. In this study, the disrupted gut barrier in rDb-Con mice promoted gut microbiota-derived LPS translocation, which accelerated systemic inflammation and resulted in higher serum levels of IL-6, TNF-$\alpha$, and IL-1$\beta$ than in the rDb-SB mice. Animal studies robustly show that peripheral inflammatory stimuli cause microglial activation in the

**FIG 6** Legend (Continued)
images of the cerebral capillary glycocalyx (scale bar = 1 $\mu$m), more detailed views of the boxed areas in the images of the cerebral capillary glycocalyx (scale bar = 500 nm), and measurements of the components of the glycocalyx syndecan-1 and HS. (d) Expression levels of tight-junction proteins ZO-1, occludin, and claudin-4 in the brains of the four groups. $n$ = 8 to 9 animals per group. Data are presented as mean values $\pm$ SD. Asterisks (*) show comparisons with the rWT-Con group, and pound signs (#) show comparisons with the rDb-Con group. *, $P < 0.05$; **, $P < 0.01$; ***, $P < 0.001$; #, $P < 0.05$; ##, $P < 0.01$; ###, $P < 0.001$.

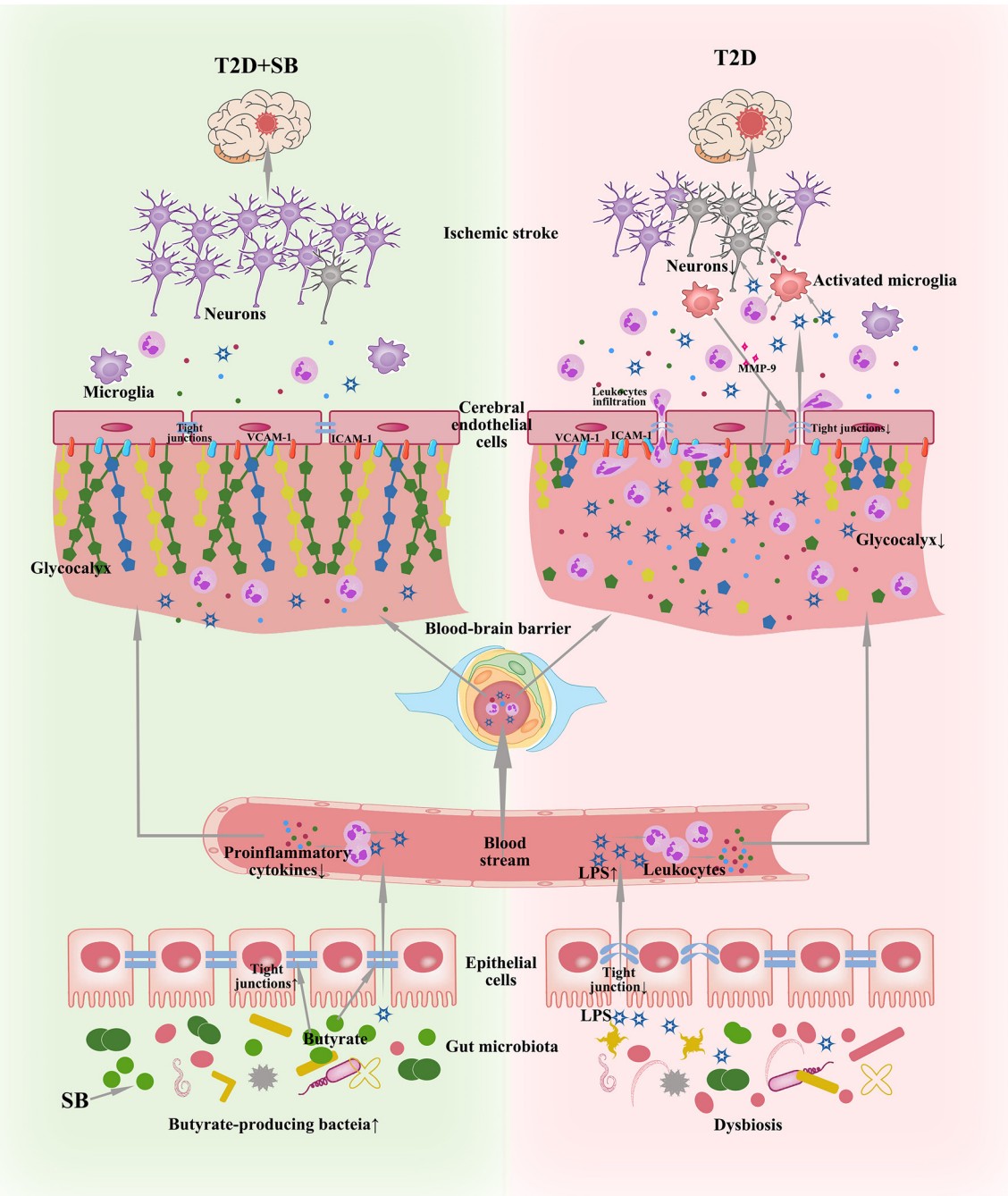

**FIG 7** SB supplementation modulates the gut microbiota of T2D, increasing the abundance of butyrate-producing bacteria with an upgraded level of butyrate, leading to a strengthened gut barrier. In the stroke context, gut-derived LPS travels through the leaky gut into the circulation, promoting the production of proinflammatory cytokines from leukocytes. The process is more prominent in T2D than in T2D+SB. The cerebral endothelial glycocalyx is the first line of defense of the BBB; it generates an exclusion zone for blood components, such as leukocytes and LPS. Adhesion molecules like VCAM-1 and ICAM-1 are harbored within the glycocalyx, so normally they do not have the chance to contact the circulating blood components. In T2D, the glycocalyx is more structurally degraded under more severe peripheral inflammation. Activated VCAM-1 and ICAM-1 are exposed and allowed to interact with leukocytes, mediating the leukocytic adhesion to the endothelial cells and migration into the brain, where they produce more inflammatory cytokines and activate the microglia, along with inflammatory cytokines and LPS that travel through the BBB. The activated microglia generate MMP-9 that destroys the BBB and glycocalyx in turn as a vicious cycle. The overt inflammatory reaction leads to more neuron death in the brain of T2D, contributing to a larger infarct volume than that in T2D supplemented with SB.

brain (32). When microglia are activated, their morphology changes in association with increased release of proinflammatory cytokines, such as IL-6, TNF-α, and IFN-γ, and the secretion of reactive oxygen and nitrogen species, leading to neuronal cell death, destruction of BBB integrity, and brain damage (33). In an *in vitro* experiment,

researchers proposed that LPS disrupts the BBB by activating microglia to damage endothelial cells (34).

Proinflammatory cytokines are transported to the brain to initiate inflammatory processes (35), leading to the extravasation of leukocytes and the upregulation of VCAM-1, ICAM-1, and MMPs (36). VCAM-1 and ICAM-1 are endothelial membrane proteins that interact with selectins on leukocytes to promote firm adhesion to the endothelium (37). In our study, the expression of these proteins, along with MMP-9, was remarkably upregulated in the brains of rDb-Con mice. MMP-9 plays a key role in the delayed opening of the BBB after ischemic stroke (38). In an IL-1$\beta$-challenged mouse model, a 5-fold increase in the level of neutrophil-derived MMP-9 in the brain was reported to be the crucial driving force in the higher rates of tight junction protein degradation after MCAO (39). In our study, we observed MMP-9 overexpression in rDb-Con mice, whereas this protein was downregulated in rDb-SB mice. The endothelial glycocalyx, a gel-like layer that covers the vascular endothelial surface and floats into the lumen of the vessels, plays a critical role in maintaining vascular integrity and cardiovascular homeostasis (40). In our previous study, we discovered that endothelial glycocalyx degradation leads to increased BBB permeability in a rat model of asphyxia cardiac arrest (41). Here, we observed that the rDb-Con group experienced the most substantial endothelial glycocalyx degradation and presented the highest serum concentrations of its components, i.e., HS and syndecan-1. However, the endothelial glycocalyx in the rDb-SB group was only mildly damaged. The degradation of endothelial glycocalyx leads to BBB disruption, aggravating ischemic stroke injury.

We acknowledge several limitations of this study. First, T2D patients had a higher body mass index (BMI) than non-T2D patients in our study and information on antidiabetic drug use was missing, both of which impact the composition of the gut microbiota greatly. Moreover, proinflammatory cytokine and LBP levels in patients should be evaluated. Second, although db/db mice present T2D-like manifestations, these manifestations are in fact secondary to genetic mutations that do not reflect disease etiology in their human counterparts. High-fat-diet-induced T2D mice would have been a more translatable T2D model for humans. Furthermore, the number of animals in the donor group was small; we should have used more mice. Third, for FMT, we used the entire fecal contents, which includes bacteria, fungi, viruses, metabolites from bacteria, and undigested food. Therefore, we did not clearly determine which component of the feces exerts effects on ischemic stroke injury. The elevated concentration of butyrate in the SB-treated group may have resulted partially from the administration of SB in the drinking water.

In conclusion, SB is sufficient to shape the gut microbiota of Db mice, which alleviates diabetes-related symptoms. Restoration of the gut microbiota of T2D mice by SB protects the BBB and reduces the cerebral infarct volume. This study provides experimental evidence that the gut microbiota of individuals with T2D can be therapeutically exploited by means of SB to protect against cerebral ischemia injury. However, future studies are required to further elucidate the underlying mechanism.

## MATERIALS AND METHODS

**Human study.** This study was conducted in the Department of Neurology of Nanfang Hospital of Southern Medical University (Guangzhou, China) from June 2017 to December 2017. The inclusion criteria of this study were as follows: (i) patients with AIS aged >18 years (ii) who were diagnosed within 3 days of stroke onset and (iii) had not yet received any AIS treatment (neither antiplatelet agents nor revascularization therapies) or had already received treatments but within 24 h. All participants provided written informed consent in accordance with the Declaration of Helsinki. This study was approved by the investigational review board of Nanfang Hospital, Southern Medical University (NFEC-2016-148), and was registered at http://www.chictr.org.cn (Chinese Clinical Trial Registry registration number ChiCTR-ROC-17011567). The diagnosis and treatment of AIS were performed according to established guidelines (42). Patients with advanced cancer, serious internal or neurological diseases, or gastrointestinal symptoms in the past 3 months were excluded. Individuals were also excluded if they had received antibiotics or probiotics within the last 3 months. Thirty-seven patients with both AIS and T2D and 37 patients with AIS but without T2D were recruited. Blood and fecal samples were collected at admission after the initial diagnosis.

**Animal study.** The experiments were approved by the Ethics Committee for Animal Care and Research of Zhujiang Hospital of Southern Medical University (Guangdong, China) and were performed according to the national guidelines (43) and the ARRIVE (Animal Research: Reporting of In Vivo Experiments) guidelines (44). Six-week-old male db/db (C57BLKS/J-m$^{+/+}$ Lepr$^{db/db}$; Db) mice were obtained from GemPharmatech Co., Ltd. (Nanjing, China), and age- and gender-matched C57BL/6J (WT) mice were purchased from Guangdong Medical Laboratory Animal Center (Guangzhou, China). All the donor mice were from the same cohort. The donor WT mice were housed with 5 animals per cage, and the Db mice were housed with 3 or 4 animals per cage. All animals were housed in an environment with controlled temperature and humidity on a 12-h:12-h light/dark cycle. After acclimatization for 1 week, the WT and Db mice were randomized to receive one of two types of drinking water for 4 weeks: (i) 0.1 mol/liter sodium chloride (NaCl, Aladdin, Shanghai, China) in distilled water as a control or (ii) 0.1 mol/liter SB (Aladdin) in distilled water. After 4 weeks of intervention, fecal samples were collected and stored in a −80°C freezer until analysis and transplantation.

For FMT, broad-spectrum antibiotics (1 g/liter ampicillin, 1 g/liter metronidazole, and 1 g/liter neomycin sulfate, Sigma-Aldrich, CA, USA) dissolved in drinking water were provided *ad libitum* to the recipient mice for 14 consecutive days. Fecal microbiota suspensions were prepared by diluting and mixing 1 g of fecal samples obtained from donor mice in 10 ml of sterile PBS, and then 0.2 ml of the suspension was intragastrically administered to each recipient mouse once daily for 14 consecutive days. There were two cohorts of recipient mice in our study. The animals used for the experiments whose results are shown in Fig. 4 and 5 were the first cohort, and the animals used for the experiments whose results are shown in Fig. 6 were the second cohort. All the recipient mice were housed with 4 or 5 animals per cage.

**Cerebral ischemia model establishment.** A cerebral ischemia model was established by inducing transient MCAO through an intraluminal suture as previously described (17). Briefly, mice were anesthetized with 1.25% tribromoethanol by intraperitoneal injection (0.02 ml/g of body weight), and the body temperature was maintained during surgery with a feedback-controlled heating pad. A monofilament was introduced into the external carotid artery and gently advanced into the internal carotid artery. After 60 min of occlusion, the filament was removed, and the ligature around the common carotid artery was removed to induce reperfusion. This surgical procedure was performed by an investigator who was blinded to the experimental groups. The exclusion criteria were as follows: (i) mice that did not survive the procedure or (ii) mice in which brain ischemia induction was not observed during the histological analysis.

**mNSS and TTC.** Neurological function was assessed using the modified neurological severity score (mNSS) (45). The test was conducted after 24 h of reperfusion. The mNSS is a composite of motor (muscle status and abnormal movement), sensory (visual, tactile, and proprioceptive sensations), and reflex tests. Neurological function was graded on a scale of 0 to 18 points (normal score, 0; maximal deficit score, 18). The higher the brain injury severity score, the more severe the injury. After neurological function was scored, the mice were anesthetized with a lethal dose of sodium pentobarbital and perfused with phosphate-buffered saline (PBS). The intact brain was dissected, cut into 2-mm tissue slices, stained with 1% triphenyl tetrazolium chloride (TTC; Sigma-Aldrich) for 15 min, and immersed in 4% formaldehyde for fixation. Twenty-four hours later, the brain slices were arranged in order and photographed. The cerebral infarct area was calculated using Image-Pro Plus software 6.0 (Fig. 2f; red areas indicate no infarction, and white areas indicate infarction). We used an edema correction formula for the infarct volume, as previously described (17). The infarct area was calculated as the area of the nonischemic hemisphere minus the noninfarcted area of the ischemic hemisphere. The total cerebral infarct volume was calculated by integrating the measured volumes of different sections in one mouse brain tissue specimen.

**16S RNA sequencing and analysis.** Bacterial genomic DNA was extracted using a MinkaGene stool DNA kit (Guangzhou, MAGIGENE) according to the manufacturer's instructions. The barcoded primers V4F (GTGYCAGCMGCCGCGGTAA) and V4R (GGACTACNVGGGTWTCTAAT) were used to amplify the 16S rRNA gene V4 variable region. PCR was performed using a previously described method (17). All PCR amplicons were mixed and sequenced using the Illumina iSeq 100 platform according to the manufacturer's protocol. The raw sequences were preprocessed and quality controlled using QIIME 2 with default parameters. A QIIME workflow script, pick_closed_reference_otus.py, was used to perform reference-based operational taxonomic unit (OTU) clustering, and USEARCH61 was used in the reference mode to search the Greengenes database version 13_8 and for BIOM (biological observation matrix) file generation. The sequences were clustered into species-level OTUs with 97% similarity. Phylogenetic relationships and the taxonomy of representative OTU sequences were determined using the Greengenes database in default mode. The Shannon index, phylogenetic diversity (PD) whole-tree index, and Chao1 index represent $\alpha$-diversity. The Shannon index indicates the number and distribution of microbial species in a sample. The PD whole-tree index indicates the range of phylogenetic distances among microbial species. The Chao1 index indicates community richness. UniFrac distances were used to analyze the $\beta$-diversity by illustrating the phylogenetic dissimilarity among samples. A smaller UniFrac distance between two samples indicates a higher similarity. As a dimensionality reduction method, PCoA was used to describe the relationships among samples based on the distance matrix and visualize the unsupervised grouping pattern of the complex data set, i.e., the microbiome. LEfSe was used to compare the discriminative data between groups. As an algorithm for high-dimensional biomarker discovery, LEfSe identifies genomic features that characterize differences between two or more biological conditions. By emphasizing statistical significance, biological consistency, and effect relevance, LEfSe determines the abundant feature with the greatest difference between conditions in accordance with biologically

meaningful categories. For each differential feature detected using LEfSe, we calculated a linear discriminant analysis value, representing the difference in the feature between groups.

**Extraction and quantification of SCFAs.** Approximately 200 mg of feces was homogenized in 1 ml of ultrapure water that contained an internal standard of 2,2-dimethylbutyric acid. The homogenate was then centrifuged at 12,000 rpm for 10 min at 4°C. The resulting supernatant was transferred to a new Eppendorf tube and mixed with 10 $\mu$l of 50% sulfuric acid, 0.5 g of sodium sulfate (Macklin, China), and 2 ml of analytically pure diethyl ether. The mixture was vortexed for 1 min and then centrifuged at 5,000 rpm for 10 min at room temperature. The ether layer was finally collected for gas chromatography with mass selective detection (5977B GC-MSD system; Agilent Technologies, Santa Clara, CA, USA). An HP-free fatty acid phase (FFAP) capillary column (30 m long, 0.25-mm internal diameter, part number 19091F433; Agilent Technologies) was used for chromatographic separation, with helium as the carrier gas. The oven temperature was increased from 90 to 180°C at a rate of 15°C/min. Gas chromatography-mass spectrometry (GC-MS) data were collected and analyzed with MassHunter Workstation software (Agilent Technologies). Final concentrations were calculated based on internal standards and are presented as micromoles per gram of wet feces ($\mu$mol/g).

**Measurements of blood lipid, fasting blood glucose, LPS, LBP, and cytokine levels.** Blood samples were collected and centrifuged at 3,000 $\times$ $g$ for 10 min, and the clear supernatant was collected. The serum levels of total cholesterol (T-CHO) and triglycerides (TGs) were quantified using kits according to the manufacturer's protocols (Nanjing Jiancheng Biotech, China). After an overnight fast, blood glucose levels in blood samples collected from the tail vein were measured using a glucometer (Accu-Chek; Roche, USA). Mouse serum LPS, LBP, DLA, IL-6, TNF-$\alpha$, and IL-1$\beta$ levels were determined using enzyme-linked immunosorbent assay (ELISA) kits (Meimian, Jiangsu, China) according to the manufacturer's protocols.

**Nissl staining and H&E staining.** Mice were anesthetized before cardiac perfusion with saline and fixation with paraformaldehyde (PFA). The brain tissue was removed carefully, postfixed with 4% PFA for 24 h, and then cryoprotected with 30% sucrose for 48 h. Serial frozen coronal sections (4 $\mu$m thick) were cut using a cryostat (Leica CM1950) at $-20$°C, and all sections containing a portion of the hippocampus were stored at $-20$°C. Then, the sections were dried in air, washed twice with distilled water (2 min/time), and stained with 1% toluidine blue for 5 min. The sections were washed 3 times with distilled water (3 min/time), placed in 70% ethanol for 2 min, washed twice with 95% ethanol (2 min/time), and then washed with xylene for 5 min. Sections were observed with a microscope (DM2500 microscope; Leica). For hematoxylin and eosin (H&E) staining, the proximal colon and ileum of mice were harvested, fixed with 4% polyformaldehyde for 24 h, and then embedded in paraffin. Next, 4-$\mu$m-thick sections were cut, dewaxed, and stained with hematoxylin and eosin using standard protocols.

**Electron microscopy and determination of glycocalyx components (syndecan-1 and HS).** The animals were perfused through the abdominal aorta with a solution containing 2.5% glutaraldehyde, 2% paraformaldehyde, and 2% lanthanum nitrate to preserve the glycocalyx in brain tissues. After the brain tissues were embedded in Epon resin, 120-nm sections were cut, and the contrast was enhanced with an incubation in 5% uranyl acetate in deionized water for 20 min and a solution containing 120 mmol/liter sodium citrate, 80 mmol/liter lead citrate, and 160 mmol/liter sodium for 2 min, the glycocalyx was photographed using a transmission electron microscope (EM 90; Zeiss, Oberkochen, Germany). Three luminal membrane segments (at least 150 nm distant from each other) of the endothelial cells of each capillary were selected in which the membrane double layer was clearly visible. Then the optical density was measured along a line perpendicular to the membrane, starting at the capillary lumen and continuing through the whole glycocalyx and the membrane double layer. The thickness of the glycocalyx was then defined as the distance between the point at the luminal side at which 50% of the maximal optical density of the glycocalyx was measured and the translucent center of the cytoplasmic membrane. Syndecan-1 concentrations were determined using an ELISA kit (Diaclone Research, Besancon, France). This kit used a solid-phase monoclonal B-B4 antibody and a biotinylated monoclonal B-D30 antibody raised against syndecan-1. The concentrations of HS were determined using an ELISA kit (Seikagaku Corporation, Tokyo, Japan), based on two antibodies specific for HS-related epitopes.

**Western blot analysis.** The ipsilateral cortex around the infarct area and the colon tissue were collected and stored at $-80$°C until further use. The brain tissues were homogenized in radioimmunoprecipitation assay (RIPA) lysis buffer (Beyotime Biotechnology, China) containing the protease inhibitor phenylmethylsulfonyl fluoride (PMSF; Beyotime Biotechnology, China) using a handheld homogenizer and incubated on ice for 20 min. The lysates were centrifuged at 12,000 $\times$ $g$ for 20 min at 4°C, and the supernatants were transferred to fresh tubes. The protein samples were resolved on a 12% or 10% SDS-PAGE gel and electrotransferred onto a polyvinylidene difluoride (PVDF) membrane. The membrane was blocked with 5% nonfat milk at room temperature for 1 h and then incubated with primary antibodies against ZO-1 (1:1,000; Abcam), occludin (1:1,000; Abcam), claudin-4 (1: 1,000; Abcam), ICAM-1 (1:1,000; Abcam), VCAM-1 (1:1,000; Abcam), MMP-9 (1:1,000; Abcam), and $\beta$-actin (1:5,000; Abcam) at 4°C overnight. The membrane was washed with Tris-buffered saline (TBS) containing 0.1% Tween 20 and incubated with a horseradish peroxidase (HRP)-conjugated goat anti-rabbit secondary antibody. The membrane was then visualized with an enhanced chemiluminescence system (Thermo Scientific, Rockford, IL). The band intensity was assessed using Image Lab software.

**Statistical analysis.** The data are presented as the mean values $\pm$ standard deviations (SD). Statistical significance of difference among the 4 groups was evaluated using one-way analysis of variance (ANOVA) with the least-significant-difference (LSD) *post hoc* test. The unpaired two-tailed Student $t$ test was performed to analyze two independent groups. SPSS 22.0 (IBM, Armonk, NY) and GraphPad Prism 5.0 (GraphPad, La Jolla, CA) software was employed for data analysis. The $\alpha$-diveristy is determined by PD

whole-tree, Shannon, Chao1, and OTUs; the $\beta$-diversity is visualized by PCoA. Nonparametric Mann-Whitney and Kruskal-Wallis tests were used to determine the statistical significance of $\alpha$-diversity measures, and permutational multivariate analysis of variance (PERMANOVA) was used to evaluate differences in $\beta$-diversity. Microbial markers with differences in abundance among the treatment groups were identified using one-way ANOVA followed by the LSD *post hoc* test for multiple comparisons and LEfSe with an LDA of >2 and *q* value of <0.05.

**Data availability.** The sequencing data have been deposited in the NCBI Sequence Read Archive (SRA) database (https://www.ncbi.nlm.nih.gov/sra) under accession number PRJNA748840.

## SUPPLEMENTAL MATERIAL

Supplemental material is available online only.

**SUPPLEMENTAL FILE 1**, PDF file, 0.4 MB.

## ACKNOWLEDGMENTS

This study was supported by the National Natural Science Foundation of China (grant number NSFC81870936), the Clinical Research Startup Program of Southern Medical University by High-level University Construction Funding of the Guangdong Provincial Department of Education (grant number LC2016PY025), and the Clinical Research Program of Nanfang Hospital, Southern Medical University (grant number 2018CR024).

J.Y., Y.H., and H.Z. designed and supervised the study; H.W., W.S. and Q.W. conducted the experiments; J.L. performed the FMT and GC-MS experiments; J.Z. provided expertise in electron microscopy and ELISA studies; X.G. and C.T. analyzed the data; H.W. drafted the paper; and Y.H. and H.Z. revised the paper. All authors read and approved the final manuscript.

We declare no conflicts of interest.

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
