## [Reviewer comments · Microbiology Spectrum]

**Microbiology
Spectrum**

Fecal transplantation from db/db mice treated with sodium butyrate attenuates ischemic stroke injury

Huidi Wang, Wei Song, Qiheng Wu, Xuxuan Gao, Jie Li, Chuhong Tan, Hong-Wei Zhou, Jiajia Zhu, Yan He, and Jia Yin

Corresponding Author(s): Jia Yin, Southern Medical University

Review Timeline:

Submission Date:	April 14, 2021
Editorial Decision:	May 25, 2021
Revision Received:	July 21, 2021
Editorial Decision:	August 4, 2021
Revision Received:	August 12, 2021
Editorial Decision:	August 15, 2021
Revision Received:	August 16, 2021
Accepted:	August 17, 2021

Editor: Steven Frese

Reviewer(s): The reviewers have opted to remain anonymous.

Transaction Report:

DOI: <https://doi.org/10.1128/Spectrum.00042-21>

May 25, 2021

Prof. Jia Yin
Southern Medical University
Neurology
Guangzhou
China

Re: Spectrum00042-21 (Modulation of the gut microbiota of type 2 diabetic mice by sodium butyrate attenuates ischemic stroke injury)

Dear Prof. Jia Yin:

Thank you for submitting your manuscript to Microbiology Spectrum. As you will see the reviewers support publication of a revised paper. Please revise the paper along the lines suggested by the reviewers. When submitting the revised version of your paper, please provide (1) point-by-point responses to the issues raised by the reviewers as file type "Response to Reviewers," not in your cover letter, and (2) a PDF file that indicates the changes from the original submission (by highlighting or underlining the changes) as file type "Marked Up Manuscript - For Review Only". Please use this link to submit your revised manuscript - we strongly recommend that you submit your paper within the next 60 days or reach out to me. Detailed information on submitting your revised paper are below.

Link Not Available

Sincerely,

Steven Frese

Journals Department
Reviewer comments:

Reviewer #1 (Comments for the Author):

Wang et al evaluate the relationship between ischaemic stroke, intestinal microbiota and type 2 diabetes in patients and mice. They show that 37 type 2 diabetic patients present with increased stroke severity compared with 37 non-diabetic patients, in association with reduced fecal butyrate levels and butyrate-producing bacteria as well as increased endotoxemia. They then show that treating hyperglycemic leptin receptor-deficient db/db mice with sodium butyrate in their drinking water for 4 weeks reduces infarct volume in response to MCA occlusion in association with reduced endotoxemia and increased fecal butyrate levels. Finally, they show that daily oral gavage of feces for two weeks from SB-treated mice to antibiotic-treated mice transfers the protection from MCA occlusion, increased fecal butyrate and suppression of endotoxemia. More in-depth analysis also revealed thickening of cerebral capillary glycocalyx, improved BBB structural integrity and reduced hippocampal inflammation and cerebral apoptosis in antibiotic-treated mice receiving feces from SB-treated mice.

The relationship between intestinal microbiota and ischaemic stroke is a novel and growing area of interest, while the relationship between intestinal microbiota and metabolic disease is more established. The authors should therefore be commended for their approach in determining the mechanistic basis of increased stroke severity in diabetes. Their study has numerous major strengths, such as the use of patients, in-depth analysis of intestinal microbiota, robust phenotyping in established animal models of diabetes and stroke and high-quality analysis of intestinal and brain tissue. Another major strength of the study is the use of FMT to show that SB improves stroke outcomes independent of its effects on glycemia. Unfortunately, there are several significant limitations in the study design which weaken the conclusions and distort the focus of the manuscript. Specifically, in Figure 2 it would have been better in this reviewer's opinion to first show in db/db mice that fecal butyrate is decreased in association with increased endotoxemia and infarct volume. This would have provided a more direct link with the clinical data in Figure 1 and also provide more evidence of the etiology of severer stroke pathology in diabetes (i.e. that of reduced SB-producing intestinal microbiota). It is also unclear whether changes in the intestinal microbiota from SB treatment or SB itself are contributing to intestinal barrier stabilization and other improvements. For example, a single inoculation of feces (rather than daily for two weeks) in antibiotic-treated mice would have provided more confidence that it is the microbiota per se doing the work. As the authors point out in their discussion (but not in their introduction) butyrate has direct barrier-stabilizing effects on enterocytes. For this reason, the title perhaps needs revising as does the titles for Figure 5 and Figure 6. Lastly, the authors do not provide the important functional evidence that disruption of the BBB in diabetes from peripheral inflammation and/or endotoxemia contributes to severer stroke outcomes.

I have the following minor comments for the authors which I hope are helpful:

Abstract, line 34: mention that C57 mice were antibiotic-treated

Introduction, line 71: would be helpful to mention direct barrier-stabilizing effects of butyrate.

Introduction, line 72: are the authors certain all of the aforementioned effects of SB are indirect through changes in the intestinal microbiota?

Introduction, line 87: would be helpful to mention how db/db mice are a model of T2D.

Table 1: Would be helpful to have fasting insulin and HbA1c values.

Figure 1: It is important to show pro-inflammatory cytokines are also lower non-T2D patients.

Additionally, LBP is a more reliable ELISA than LPS so would be helpful to have these values too.

Figure 2: Was barrier protein expression in the ileum and colon analyzed in these mice too?

It would be helpful to have a summary figure of the proposed mechanism for severer stroke in

diabetes and resolution by SB treatment. This is especially because the discussion proposes different distinct mechanisms such as roles for proinflammatory cytokines in activating microglia and BBB disruption as well as a direct effect of LPS.

Discussion, line 269: gut barrier integrity was not assessed before stroke in this study so it cannot be concluded what comes first.

Reviewer #2 (Comments for the Author):

In this study the authors describe an interesting concept that butyrate reduces brain injury via the gut microbiota. They provide clinical data and several mouse experiments that provide evidence for this concept. However, I have a few concerns about the translatability to humans.

Table 1: Body weight and BMI are missing, which influence several factors such as the gut microbiota and blood glucose. Further, anti-diabetic drugs are not mentioned that also have an influence on the gut microbiota.

Line 100: This second part of the sentence is redundant.

Line 102-105: This part is not necessary in the result section, more methods.

Figure 1d: The letter "d" is missing in the figure.

Figure 2: The authors report 5-10 animals per group. That is rather low for an animal experiment focusing on the gut microbiota. Are those animals housed singular (coprophagia) and did they repeat the experiments with another cohort?

Figure 2: Db/db is a very sick mouse model, diet induced obese mice would have been more appropriate and relatable to the human situation.

Figure 2f: Major portions of butyrate is absorbed in the intestine, does it end up in the brain or is the effect mainly by improving the intestinal barrier function?

Figure 3: Wt and db mice were ordered from two different providers/litters, therefore statistical analysis on the gut microbiota might not be appropriate.

Figure 3b: The figures are difficult to read (pixelated)

Figure 3e: As mentioned before, the db/db mouse model is an extreme model which is almost not translatable to humans. Here, wild-type mice improve in intestinal SCFAs concentrations but not in clinical parameters (Figure 2). Whether these findings can be translated to humans is questionable.

Figure 4: Please indicate how many cohort were used here. Particularly findings about gut microbiota are dependent on the litter.

Line 153: "markedly" is a strong word for such a small reduction.

Figure S1: How much does the microbiota contribute the clinical parameters of T2D? SCFAs increase (particularly in wt) but clinical parameters are not changing after FMT.

Figure 5: Again, 4-6 mice is very low. Different cohort? Single housed?

Line 207: Improved stroke outcomes were only significant in db/db mice, not in the wt mice. Therefore it is not "independent" of T2D background.

Line 214: Major confounding factors are obesity and anti-diabetic drug usage. The authors cannot conclude that only T2D is associated with lower butyrate bacteria.

Line 228: Clinical evidence is still lacking of butyrate supplementation in T2D. It is a great concept, but did not improve diabetes in human studies. Only rodent studies show improvements.

Line 242: Why did the authors transplant mouse feces instead of human feces. That would have more clinical impact.

Line 280: Does LPS end up in the brain to induce inflammation/damage?

The discussion is very long. In my opinion it can be shortened at various parts.

Line 309: Sample size is not the major concern here. See above. Please expand limitations according to obesity/anti-diabetic drugs.

Line 311: db/db develop T2D, but are barely translatable to humans. High fat diet treated mice are more translatable.

Staff Comments:

Preparing Revision Guidelines

For complete guidelines on revision requirements, please see the Instructions to Authors at [link to page]. **Submissions of a paper that does not conform to Microbiology Spectrum guidelines will delay acceptance of your manuscript.**

Please return the manuscript within 60 days; if you cannot complete the modification within this time period, please contact me. If you do not wish to modify the manuscript and prefer to submit it to another journal, please notify me of your decision immediately so that the manuscript may be formally withdrawn from consideration by Microbiology Spectrum.

If you would like to submit an image for consideration as the Featured Image for an issue, please contact Spectrum staff.

Reviewer comments:

Reviewer #1 (Comments for the Author):

(1). The relationship between intestinal microbiota and ischaemic stroke is a novel and growing area of interest, while the relationship between intestinal microbiota and metabolic disease is more established. The authors should therefore be commended for their approach in determining the mechanistic basis of increased stroke severity in diabetes. Their study has numerous major strengths, such as the use of patients, in-depth analysis of intestinal microbiota, robust phenotyping in established animal models of diabetes and stroke and high-quality analysis of intestinal and brain tissue. Another major strength of the study is the use of FMT to show that SB improves stroke outcomes independent of its effects on glycemia. Unfortunately, there are several significant limitations in the study design which weaken the conclusions and distort the focus of the manuscript.

Response: We deeply appreciate your careful review and constructive comments. We have studied these comments carefully and made corrections point by point with yellow mark in the manuscript which we hope meet with your approval.

(2). Specifically, in Figure 2 it would have been better in this reviewer's opinion to first show in db/db mice that fecal butyrate is decreased in association with increased endotoxemia and infarct volume. This would have provided a more direct link with the clinical data in Figure 1 and also provide more evidence of the etiology of severer stroke pathology in diabetes (i.e. that of reduced SB-producing intestinal microbiota).

Response: We thank the reviewer for your helpful and insightful suggestions. We have added the correlation analysis in Figure 2g and made the modification in the Result section accordingly. Line 138-139, “A significant negative correlation was noted between fecal butyrate and LPS and infarct volume (Fig. 2g)”.

(3). It is also unclear whether changes in the intestinal microbiota from SB treatment or SB itself are contributing to intestinal barrier stabilization and other improvements. For example, a single inoculation of feces (rather than daily for two weeks) in

antibiotic-treated mice would have provided more confidence that it is the microbiota per se doing the work. As the authors point out in their discussion (but not in their introduction) butyrate has direct barrier-stabilizing effects on enterocytes. For this reason, the title perhaps needs revising as does the titles for Figure 5 and Figure 6.

Response: We are very grateful for your insightful comment, which helps us make a further improvement for our work. Indeed, a single inoculation of feces would provide more convincing evidence that it is the gut microbiota that makes the difference. Therefore, we have revised the title of the manuscript. The new title is “Fecal transplantation from db/db mice treated with sodium butyrate attenuates ischemic stroke injury”, and the new titles for Figure 5 and Figure 6 are “Fecal transplantation from db/db mice treated with SB attenuates ischemic stroke injury and gut barrier destruction” and “Fecal transplantation from db/db mice treated with SB attenuates ischemic stroke injury by protecting the BBB”, respectively.

(4). Lastly, the authors do not provide the important functional evidence that disruption of the BBB in diabetes from peripheral inflammation and/or endotoxemia contributes to severer stroke outcomes.

Response: We are sorry for not being able to clearly clarify the mechanism by which BBB disruption from peripheral inflammation leads to severer stroke outcomes. So we add a summary figure to better display the possible mechanism in Figure 7 which we hope would be helpful.

(5). Abstract, line 34: mention that C57 mice were antibiotic-treated.

Response: We apologize for the negligence and the sentence has been corrected. Line 35 “Fecal samples were collected from T2D mice after SB or NaCl treatment and were transplanted into antibiotic-treated C57BL/6 mice”.

(6). Introduction, line 71: would be helpful to mention direct barrier-stabilizing effects of butyrate.

Response: Thank you for your kind suggestion and we have removed the information

concerning the barrier-stabilizing effects of butyrate from the discussion section to the Introduction section. Line 73-76 “Moreover, butyrate increases mitochondrial-dependent oxygen consumption in enterocytes, stabilizes the hypoxia-inducible factor (HIF) that is involved in barrier protection, and up-regulates the expression of HIF-target genes that increase barrier function”.

(7) Introduction, line 72: are the authors certain all of the aforementioned effects of SB are indirect through changes in the intestinal microbiota?

Response: Thanks for your careful comment. We have gone through the references again and we apologize for the misinformation, we think it is the best that we remove the sentence because of the redundancy. And we added the information concerning the barrier-stabilizing effects of butyrate in here. Thank you again for your good suggestion.

(8). Introduction, line 87: would be helpful to mention how db/db mice are a model of T2D.

Response: Thank you for your kind suggestion and we have added the information in line 91-93 “Due to single-gene mutations that lead to deactivation by the cognate receptor of the satiety factor leptin, db/db mice spontaneously develop severe hyperphagia leading to obesity and manifestation of some T2D-like characteristics”.

(9). Table 1: Would be helpful to have fasting insulin and HbA1c values.

Response: Thanks for pointing out, we have added HbA1c values in Table 1. Unfortunately, fasting insulin is not the routine test in our department, so we don't have the relative data, we seek for your understanding.

(10). Figure 1: It is important to show pro-inflammatory cytokines are also lower non-T2D patients. Additionally, LBP is a more reliable ELISA than LPS so would be helpful to have these values too.

Response: It is true as you suggested that it would be more convincing to show

pro-inflammatory cytokines and LBP level in the patients. Regrettably, these tests were not within our study design when we conceived the study, and we didn't have any blood sample left for further tests. We are deeply sorry and seek for your understanding.

(11). Figure 2: Was barrier protein expression in the ileum and colon analyzed in these mice too?

Response: Thanks for your question and we analyzed the barrier expression in the intestine. As is shown in Figure 2h, we observed higher expression of ZO-1, Occludin and Claudin-4 in Db-SB mice than in Db-Con mice

(12). It would be helpful to have a summary figure of the proposed mechanism for severer stroke in diabetes and resolution by SB treatment. This is especially because the discussion proposes different distinct mechanisms such as roles for proinflammatory cytokines in activating microglia and BBB disruption as well as a direct effect of LPS.

Response: We appreciate for your warm comment. We have added a summary figure (Figure 7) indicating the mechanism for severer stroke in diabetes which could be attenuated by SB treatment. We hope it would be helpful.

(13). Discussion, line 269: gut barrier integrity was not assessed before stroke in this study so it cannot be concluded what comes first.

Response: Thanks for your insightful comment. The sentence has been removed for better clarity.

Reviewer #2 (Comments for the Author):

(1). In this study the authors describe an interesting concept that butyrate reduces brain injury via the gut microbiota. They provide clinical data and several mouse experiments that provide evidence for this concept. However, I have a few concerns about the translatability to humans.

Response: We really appreciate your efforts and comments on our manuscript. We have revised our manuscript point by point according to your comments and suggestions with yellow mark in the manuscript which we hope meet with your approval.

(2). Table 1: Body weight and BMI are missing, which influence several factors such as the gut microbiota and blood glucose. Further, anti-diabetic drugs are not mentioned that also have an influence on the gut microbiota.

Response: Thank you for your kind suggestion and we have added the information of body weight ($p=0.096$) and BMI ($p=0.025$) in the Table 1. Indeed, anti-diabetic drugs do influence the composition of the gut microbiota. Regrettably, we didn't collect the all the information concerning the use of the anti-diabetic drug of the T2D patients, so the data are incomplete. It is our negligence and we will keep it in mind when we design the study next time.

(3). Line 100: This second part of the sentence is redundant.

Response: Thank you for your suggestion. We have removed the second part of the sentence.

(4). Line 102-105: This part is not necessary in the result section, more methods.

Response: Thanks for your suggestion. We have removed this part from result section in the methods section.

(5). Figure 1d: The letter "d" is missing in the figure.

Response: We are deeply sorry for the mistake we made. The letter “d” has been added in the Figure 1.

(6). Figure 2: The authors report 5-10 animals per group. That is rather low for an animal experiment focusing on the gut microbiota. Are those animals housed singular (coprophagia) and did they repeat the experiments with another cohort?

Response: Thanks for your insightful comments. Indeed, the amount of the animals per group is small, it is a limitation of our study. We have added the information in line 373 “All the donor mice were from the same cohort. The donor Wt mice were housed 5 animals per cage and the Db mice were house 3-4 animals per cage. All animals were housed in an environment with controlled temperature and humidity on a 12-h:12-h light/dark cycle” and in line 388 “There were two cohorts of recipient mice in our study, the animals used in figure 4 and 5 were the first cohort, the animals used in figure 6 were the second cohort, all the recipient mice were house 4-5 animals per cage”.

(7). Figure 2: Db/db is a very sick mouse model, diet induced obese mice would have been more appropriate and relatable to the human situation.

Response: Thanks for your comments. It is true that db/db is a very sick mouse model of T2D and diet-induced obese mice would have better translatability to humans. However, we were concerned that diet especially the high fat diet would have great impact on the composition of the gut microbiota, so it is hard to tell whether the changed gut microbiota was resulted from high fat diet or the disease model. Moreover, undigested fat in the feces would be transferred into the recipient mice during FMT, which might incur metabolic syndrome in recipient mice. Therefore, we hope that you understand the predicament we were in. Thanks again for your kind suggestions.

(8). Figure 2f: Major portions of butyrate is absorbed in the intestine, does it end up in

the brain or is the effect mainly by improving the intestinal barrier function?

Response: Thanks for your question. Studies using isotope-labelled butyrate were able to demonstrate carrier-mediated uptake of butyrate and other monocarboxylates into the brain (PMID: 4712154, 758224), although more recently this capacity was demonstrated to be limited (PMID: 23906667). Importantly, in order to affect brain function, butyrate does not necessarily need to enter the brain but can also indirectly influence process in the brain by stimulating for example the peripheral nervous system, regulating immune system function, or improving the intestinal function. Therefore, butyrate might affect brain function in various ways. Because butyrate is rapidly metabolized in the intestine, so we applied the FMT technique to minimize the effects of long-term butyrate supplement.

(9). Figure 3: Wt and db mice were ordered from two different providers/litters, therefore statistical analysis on the gut microbiota might not be appropriate.

Response: Thanks for your valuable comments and we are sorry for this mistake. As Guangdong Medical Lab Animal Center doesn't provide db/db mice, we had to purchase from GemPharmatech Co, Ltd. Indeed, mice from different providers might present different gut microbiota, so we tried to minimize the effect by subjecting mice to acclimatization for 1 week before any intervention. Moreover, all the mice were raised under the same condition.

(10). Figure 3b: The figures are difficult to read (pixelated)

Response: We are very sorry for the inconvenience we caused in your reading. We have improved the quality of the figure 3b and figure 4c.

(11). Figure 3e: As mentioned before, the db/db mouse model is an extreme model which is almost not translatable to humans. Here, wild-type mice improve in intestinal SCFAs concentrations but not in clinical parameters (Figure 2). Whether these findings can be translated to humans is questionable.

Response: Thanks for your valuable comment. Indeed, Wt-SB mice possessed a

higher concentration of butyrate and it should lead to a smaller infarct volume than Wt-Con mice. However, the cerebral protective effects might be context-dependent. For example, in a study by Spychala et al. (PMID: 29733457), gut microbiota from young (8–12 weeks, with high concentrations of SCFAs) or aged (18–20 months, with low concentrations of SCFAs) mice were transplanted into young or aged mice before MCAO. They found significant differences in cerebral infarct volume between aged mice that received young microbiota compared to aged mice that received aged microbiota, whereas no significant differences were found between young mice that received young microbiota compared to young mice that received aged microbiota. It indicates that the beneficial effects of butyrate might be more pronounced in individuals with bad physical conditions. Moreover, in a study by Kim et al. (PMID: 17371805), different doses of SB were injected into a rat model of MCAO, they found that the cerebral protective effect was dose-dependent with an approximately 50% decrease in infarct volume at the dose of 200 to 300 mg/kg, whereas the protective effect was less pronounced at higher doses, notably 700 mg/kg, which induced only a 22% decrease. Interestingly, a recent study (PMID: 33420074) found that excess butyrate in the context of liver cancer is hazardous because it obstructs the immune system from functioning. Therefore, butyrate might exert more beneficial effects on T2D individuals that were already morbidly short of butyrate. To healthy individuals that have a physiological level of butyrate, the effects might be less pronounced.

(12). Figure 4: Please indicate how many cohort were used here. Particularly findings about gut microbiota are dependent on the litter.

Response: Thanks for your comment. There were two cohorts of recipient mice in our study, the mice used in Figure 4 and Figure 5 were from the same cohort, the feces were collected on the same date under the same condition. The mice that were used in Figure 6 were from another cohort. We have added the information in the Methods section line 373 and line 388.

(13). Line 153: "markedly" is a strong word for such a small reduction.

Response: We apologize for the overstatement. The sentence has been corrected, line 163 “The Chao1 index of α -diversity was decreased in mice that received fecal microbiota from NaCl-treated Db mice (rDb-Con)”.

(14). Figure S1: How much does the microbiota contribute the clinical parameters of T2D?

Response: Thanks for your kind comment. Several studies have shown that the gut microbiota might be an important contributor to the development of T2D. Ridaura et al. (PMID: 24009397) discovered that the obesity-associated metabolic phenotype can be transmitted from humans to mice by FMT. In males with metabolic syndrome, FMT from lean male donors results in a significant improvement in insulin sensitivity, increased gut microbiota diversity and a remarkable increase in butyrate-producing bacteria (PMID: 22728514). In a recent study, Yu et al. (PMID: 33223509) discovered that gut microbiota transplantation from db/db mice induces diabetes-like phenotypes such as blood glucose, body weight, food and water intake in pseudo germ-free mice.

(15). SCFAs increase (particularly in wt) but clinical parameters are not changing after FMT.

Response: Thanks for your kind comment. To our knowledge, the effects of butyrate on diabetes-related parameter are less pronounced in wild-type mice. So it is possible that the clinical parameters do not change along with SCFAs levels in wild-type mice that are not lack of butyrate. In db/db mice, we speculate that the concentration of fecal butyrate tested in our study is resulted from a dynamic balance between the production from bacteria and the absorption into the epithelial cells. So the anti-diabetic effects of butyrate observed in donor db/db mice might be from the exogenous butyrate supplement.

(16). Figure 5: Again, 4-6 mice is very low. Different cohort? Single housed?

Response: We apologized for the mistake we made. The amount of mice used in Figure 5 is 14-16 per group, they were from the same cohort. And the mice used in

Figure 6 were from another cohort. We've conducted two independent experiments for the recipient mice. All the recipient mice were house 4-5 per cage.

(17). Line 207: Improved stroke outcomes were only significant in db/db mice, not in the wt mice. Therefore it is not "independent" of T2D background.

Response: We are sorry for the mistake and we have corrected the sentence in line 221 "The primary finding of the present study is that SB modulates the gut microbiota profile of T2D mice and increases the fecal butyrate concentration, which improves stroke outcomes".

(18). Line 214: Major confounding factors are obesity and anti-diabetic drug usage. The authors cannot conclude that only T2D is associated with lower butyrate bacteria.

Response: Thanks for your comment and we have removed the sentence.

(19). Line 228: Clinical evidence is still lacking of butyrate supplementation in T2D. It is a great concept, but did not improve diabetes in human studies. Only rodent studies show improvements.

Response: Thanks for your comment and we have removed the improper description.

(20). Line 242: Why did the authors transplant mouse feces instead of human feces. That would have more clinical impact.

Response: Thanks for your insightful comment. The reason why we chose the mouse feces is that we were worried that there were too many confounding factors accompanied with human feces, such as diet, region, exercise, drugs, comorbidities, smoking, alcohol and so on, which might partly explain why there were no differences in α - or β -diversity in AIS patients with or without T2D. But we are convinced that if the results could be replicated using human feces, it would certainly have more clinical impact.

(21). Line 280: Does LPS end up in the brain to induce inflammation/damage?

Response: Thanks for your comment. In fact, LPS could end up in the brain to induce inflammation. The BBB is disrupted in ischemic stroke, and is sensitized to further disruptive changes by systemic inflammation (PMID: 28821274). Not only is LPS accountable for BBB disruption, it can cross the BBB to access brain compartment, causing increased microglial IL-1 α expression and marked granulocyte recruitment throughout the ipsilateral hemisphere (PMID: 22114895). We have added a summary figure of the proposed mechanism in Figure 7, which we hope would be helpful.

(22). The discussion is very long. In my opinion it can be shortened at various parts.

Response: Thanks for your kind suggestion. We have made substantial revision in the discussion section and removed the redundant part.

(23). Line 309: Sample size is not the major concern here. See above. Please expand limitations according to obesity/anti-diabetic drugs.

Response: Thanks for your kind suggestion. We have corrected the limitations in the discussion which we hope would meet with your approval. Line 326 “First, T2D patients have a higher BMI index than non-T2D patients in our study and the information of anti-diabetic drug use are missing, both of which impacts the composition of the gut microbiota greatly”.

(24). Line 311: db/db develop T2D, but are barely translatable to humans. High fat diet treated mice are more translatable.

Response: We are truly appreciative of your suggestion. Indeed, high fat diet induced T2D mice are more translatable to humans, we have corrected the improper description in the discussion. Line 332 “Second, although db/db mice present T2D-like manifestations, these manifestations are in fact secondary to genetic mutations that do not reflect disease etiology in their human counterparts. High fat diet-induced T2D mice would have been a more translatable T2D model for humans”.

August 4, 2021

Prof. Jia Yin
Southern Medical University
Neurology
Guangzhou
China

Re: Spectrum00042-21R1 (Fecal transplantation from db/db mice treated with sodium butyrate attenuates ischemic stroke injury)

Dear Prof. Jia Yin:

Thank you for your detailed response to the Reviewer's queries.

Please add a response to Reviewer #1's comment (#10) regarding LBP and serum cytokines as a limitation of the study and a note addressing the comment regarding Reviewer #2's comment #6 on the N of the study as a limitation, as you have noted in the Response to Reviewers.

Thank you for submitting your manuscript to Microbiology Spectrum. As you will see your paper is very close to acceptance. Please modify the manuscript along the lines I have recommended. As these revisions are quite minor, I expect that you should be able to turn in the revised paper in less than 30 days, if not sooner. If your manuscript was reviewed, you will find the reviewers' comments below.

When submitting the revised version of your paper, please provide (1) point-by-point responses to the issues I raised in your cover letter, and (2) a PDF file that indicates the changes from the original submission (by highlighting or underlining the changes) as file type "Marked Up Manuscript - For Review Only". Please use this link to submit your revised manuscript. Detailed information on submitting your revised paper are below.

Link Not Available

Sincerely,

Steven Frese

Reviewer comments:

Preparing Revision Guidelines

- point-by-point responses to the issues I raised in your cover letter
- Upload a compare copy of the manuscript (without figures) as a "Marked-Up Manuscript" file.
- Each figure must be uploaded as a separate file, and any multipanel figures must be assembled into one file.
- Manuscript: A .DOC version of the revised manuscript
- Figures: Editable, high-resolution, individual figure files are required at revision, TIFF or EPS files are preferred

For complete guidelines on revision requirements, please see the Instructions to Authors at [link to page]. **Submissions of a paper that does not conform to Microbiology Spectrum guidelines will delay acceptance of your manuscript.**

Please return the manuscript within 60 days; if you cannot complete the modification within this time period, please contact me. If you do not wish to modify the manuscript and prefer to submit it to another journal, please notify me of your decision immediately so that the manuscript may be formally withdrawn from consideration by Microbiology Spectrum.

If you would like to submit an image for consideration as the Featured Image for an issue, please contact Spectrum staff.

(1). Please add a response to Reviewer #1's comment (#10) regarding LBP and serum cytokines as a limitation of the study.

Response: Thanks for your kind suggestion. We have added the sentence in line 329-330 "Moreover, pro-inflammatory cytokines and LBP levels in patients should be evaluated".

(2). Please add a note addressing the comment regarding Reviewer #2's comment #6 on the N of the study as a limitation, as you have noted in the Response to Reviewers.

Response: We apologize for the negligence and we have added the sentence in line 338-339 "Furthermore, the amount of animals in the donor group is small, we should've adopted more mice".

August 15, 2021

Prof. Jia Yin
Southern Medical University
Neurology
Guangzhou
China

Re: Spectrum00042-21R2 (Fecal transplantation from db/db mice treated with sodium butyrate attenuates ischemic stroke injury)

Dear Prof. Jia Yin:

Thank you for submitting your manuscript to Microbiology Spectrum. As you will see your paper is very close to acceptance. Please modify the manuscript along the lines I have recommended. As these revisions are quite minor, I expect that you should be able to turn in the revised paper in less than 30 days, if not sooner. You will find the Editorial comments below.

When submitting the revised version of your paper, please provide (1) point-by-point responses to the issues I raised in your cover letter, and (2) a PDF file that indicates the changes from the original submission (by highlighting or underlining the changes) as file type "Marked Up Manuscript - For Review Only". Please use this link to submit your revised manuscript. Detailed information on submitting your revised paper are below.

Link Not Available

Sincerely,

Steven Frese

Editorial comments:

Thank you for responding to the Reviewer's queries. To meet the Journal's requirements on data availability, please deposit the sequencing data in one of the listed repositories: <https://journals.asm.org/list-data-repositories> and note this in the manuscript file.

Preparing Revision Guidelines

- point-by-point responses to the issues I raised in your cover letter
- Upload a compare copy of the manuscript (without figures) as a "Marked-Up Manuscript" file.
- Each figure must be uploaded as a separate file, and any multipanel figures must be assembled into one file.
- Manuscript: A .DOC version of the revised manuscript
- Figures: Editable, high-resolution, individual figure files are required at revision, TIFF or EPS files are preferred

For complete guidelines on revision requirements, please see the Instructions to Authors at [link to page]. **Submissions of a paper that does not conform to Microbiology Spectrum guidelines will delay acceptance of your manuscript.**

Please return the manuscript within 60 days; if you cannot complete the modification within this time period, please contact me. If you do not wish to modify the manuscript and prefer to submit it to another journal, please notify me of your decision immediately so that the manuscript may be formally withdrawn from consideration by Microbiology Spectrum.

If you would like to submit an image for consideration as the Featured Image for an issue, please contact Spectrum staff.

(1) To meet the Journal's requirements on data availability, please deposit the sequencing data in one of the listed repositories: <https://journals.asm.org/list-data-repositories> and note this in the manuscript file.

Response: Thanks for reminding us. We have deposited the sequencing data in the NCBI Sequence Read Archive (SRA). We have noted this in the manuscript file line 543-544 “The sequencing data have been deposited in the NCBI Sequence Read Archive (SRA) database (<https://www.ncbi.nlm.nih.gov/sra>) under accession number PRJNA748840”.

August 17, 2021

Prof. Jia Yin
Southern Medical University
Neurology
Guangzhou
China

Re: Spectrum00042-21R3 (Fecal transplantation from db/db mice treated with sodium butyrate attenuates ischemic stroke injury)

Dear Prof. Jia Yin:

Your manuscript has been accepted, and I am forwarding it to the ASM Journals Department for publication. You will be notified when your proofs are ready to be viewed.

Sincerely,

Steven Frese
Editor, Microbiology Spectrum
